# The Multifaceted Role of Neutrophils in NSCLC in the Era of Immune Checkpoint Inhibitors

**DOI:** 10.3390/cancers16142507

**Published:** 2024-07-10

**Authors:** Shucheng Miao, Bertha Leticia Rodriguez, Don L. Gibbons

**Affiliations:** 1Department of Thoracic Head & Neck Medical Oncology, MD Anderson Cancer Center, Houston, TX 77030, USA; smiao@mdanderson.org (S.M.); blrodriguez@mdanderson.org (B.L.R.); 2The University of Texas MD Anderson Cancer Center, UTHealth at Houston Graduate School of Biomedical Sciences, Houston, TX 77030, USA

**Keywords:** neutrophils, non-small-cell lung cancer, tumor microenvironment, immunotherapy

## Abstract

**Simple Summary:**

Lung cancer is the leading cause of cancer-related death in the U.S. and the majority of these cases are non-small-cell lung cancer (NSCLC). While immune checkpoint inhibitors (ICIs) are initially effective, patients often develop resistance to the treatment. Tumor-associated neutrophils (TANs) play a key role in the tumor environment and immune response. The main challenge is to understand how TANs influence the effectiveness of ICI treatment. This review discusses current research on how neutrophils interact with cancer cells and other components in the tumor environment. Ongoing clinical trials are exploring treatments that target these neutrophils to improve ICI therapy in NSCLC.

**Abstract:**

Lung cancer is the most common cause of cancer-related death in both males and females in the U.S. and non-small-cell lung cancer (NSCLC) accounts for 85%. Although the use of first- or second-line immune checkpoint inhibitors (ICIs) exhibits remarkable clinical benefits, resistance to ICIs develops over time and dampens the efficacy of ICIs in patients. Tumor-associated neutrophils (TANs) have an important role in modulating the tumor microenvironment (TME) and tumor immune response. The major challenge in the field is to characterize the TANs in NSCLC TME and understand the link between TAN-related immunosuppression with ICI treatment response. In this review, we summarize the current studies of neutrophil interaction with malignant cells, T-cells, and other components in the TME. Ongoing clinical trials are aimed at utilizing reagents that have putative effects on tumor-associated neutrophils, in combination with ICI. Elevated neutrophil populations and neutrophil-associated factors could be potential therapeutic targets to enhance anti-PD1 treatment in NSCLC.

## 1. Introduction

Lung cancer is the second most common cancer in each sex and the leading cause of cancer death in the USA, making up approximately 25% of all cancer-related deaths [1]. Approximately 85% of lung cancers are identified as non-small-cell lung cancer (NSCLC) [2]. The three major subtypes of NSCLC are adenocarcinoma (40%), squamous cell carcinoma (30%), and large-cell carcinoma (10%) [3]. Based on stage, histology, genetic mutation, and patient conditions, the NSCLC treatment regime includes surgery, chemotherapy, radiotherapy, molecular targeted therapy, and/or immunotherapy [1]. Compared to or in combination with conventional treatments, immune checkpoint inhibitors (ICIs) such as PD-1/PD-L1- and CTLA-4-blocking antibodies have transformed the treatment paradigm for NSCLC and demonstrated a significant survival benefit in patients [1,2]. Although ICIs exhibit remarkable clinical benefits compared to chemotherapy, the majority of NSCLC patients are refractory or develop resistance to ICIs over time [4]. The basis for the development of resistance to ICIs remains to be fully elucidated and there is an urgent need to clarify the underlying cellular and molecular mechanisms.

Recent studies have concluded that the failure of ICI treatment is likely due to the absence or inhibition of antigen-driven immune response or immune-suppressive components in the tumor microenvironment (TME) that abolish active immune response [5,6,7]. The TME consists of an extracellular matrix, blood vessels, immune cells, stromal cells, fibroblasts, and signaling molecules [8], all essential components in understanding cancer progression and response to therapeutics [9,10]. Among tumor-infiltrating immune cells in the TME, neutrophils are the most prevalent immune cells found in NSCLC, accounting for about 20% of CD45^+^ cells [11,12].

Neutrophils, also known as polymorphonuclear granulocytes, play a key role in the activation and regulation of both the innate and adaptive immune systems. They have long been known to serve as an essential line of defense against infectious agents. In NSCLC, tumor-associated neutrophils (TANs) have emerged as an important component of the TME that negatively correlates with the prognosis and clinical benefits of immunotherapy [13]. The neutrophil-to-lymphocyte ratio (NLR) has been identified as an adverse prognostic indicator in stage III-IV NSCLC patients treated with immunotherapy [2,14]. An elevated pretreatment NLR can predict diminished therapeutic response to ICIs [14,15]. However, the underlying mechanism of their utility as a prognostic or predictive biomarker is still undefined. The elucidation of the basis of this link might shed light on the investigation of novel combination therapies. In this review, we will discuss the multifaceted role of neutrophils in the TME, including their interaction with malignant cells and T-cells, and targeting tumor-associated neutrophils as a potential therapeutic approach to augment response to immune checkpoint inhibitors.

## 2. TAN Heterogeneity in Tumors and Correlation with ICI Response

The life cycle of neutrophils begins with differentiation in the bone marrow (BM). Neutrophils are released into the vasculature and account for 50–70% of all peripheral leukocytes in humans and 10–25% in mice [16]. They are the first responders to infection and inflammation and are rapidly recruited to tissue sites. In contrast to prior assumptions that neutrophils have a short lifespan (about 19 h), recent studies have shown their lifespan may increase after activation [17]. Mature neutrophils contain granules filled with lysozyme, defensins, metalloproteases (MMPs), myeloperoxidase (MPO), neutrophil elastase (NE), and enzymes that trigger oxidative stress to exert their cytotoxic effects [14,18,19,20].

Most inflammatory cells in solid tumors are neutrophils and they are associated with multiple components of tumor progression including primary tumor proliferation, the formation of metastatic niches, the progression of metastases, immune suppression, and cancer stem cell maintenance [21]. The prevalence of TANs consistently correlates with poor prognosis and poor response to therapies in various cancer types [22]. In mouse models, upon recruitment to the tumor, tumor-associated neutrophils (TANs) can undergo polarization influenced by tumor microenvironmental factors, resulting in two distinct phenotypes: anti-tumor N1 TANs and tumor-promoting N2 TANs. N1 TANs possess a highly cytotoxic nature, exhibiting a short lifespan and mature phenotypes that contribute to the enhancement of cytotoxic T-cell responses [20]. In contrast, N2 TANs display low cytotoxicity, have a longer lifespan, exhibit immature phenotypes, and exert immunosuppressive activity [20]. The conversion of TANs to pro-tumor phenotypes in mice, namely N2 TANs, is primarily driven by TGFβ, while type 1 interferon (IFN) can shift TANs towards an anti-tumor phenotype [23,24]. Consequently, the cytokine milieu plays a critical role in shaping and sustaining the presence of N1 or N2 TANs within the tumor microenvironment in animal models. Transcriptomic studies in mice have also revealed distinct gene expression patterns between N1 and N2 TANs, highlighting alterations in structural genes, stress responses, granule proteins, vesicle formation, apoptosis, inflammatory response, cytokine signaling, and antigen presentation pathways [25].

Myeloid-derived suppressor cells (MDSCs) are a group of heterogeneous cells derived from myeloid progenitor cells. There are two subgroups of MDSCs: monocytic-MDSCs (m-MDSCs) with monocyte phenotypes and polymorphonuclear-MDSCs (PMN-MDSCs), also known as granulocytic-MDSCs (g-MDSCs), with neutrophil-like phenotypes [26,27,28,29,30]. In mice, PMN-MDSCs/g-MDSCs are typically CD11b^+^Ly6G^+^Ly6C^low^ while m-MDSCs are CD11b^+^Ly6G^-^Ly6C^high^ [26]. In humans, PMN-MDSC/g-MDSCs are identified as CD11b^+^CD66b^+^CD15^+^CD14^-^CD33^low^, and m-MDSCs are distinguished by CD33^+^ [26,31]. The states of MDSC were highly conserved between mice and humans [32]. The gene signature of patient tumor-associated PMN-MDSCs resembles that of activated PMN-MDSCs in tumor-bearing mice. Importantly, this gene signature is strongly associated with worse clinical outcomes in cancer patients [33]. Single-cell RNA sequencing analysis demonstrated that PMN-MDSCs have both mature and immature neutrophil gene signatures. Veglia and colleagues reported that one of the PMN-MDSCs subclusters showed a distinct gene enrichment pattern, with an abundance of genes associated with immature neutrophils including Ngp, Ltf, Cd177, Anxa1, S100a8, S100a9, Cebpe, Ltb4r1, and Cybb. Another PMN-MDSCs subcluster exhibited a gene profile similar to more mature neutrophils, with a higher expression of genes in chemotaxis, inflammation, and NO and ROS signaling pathways [33].

MDSCs are also highly plastic. M-MDSCs can not only differentiate into macrophages or dendritic cells but can also switch to the g-MDSC phenotype [28,29,34,35,36]. There is ongoing debate and controversy regarding the distinctions between N2 neutrophils and g-MDSCs. While some studies suggest that the term g-MDSC should apply to all neutrophils with pro-tumor phenotypes (N2) found in the TME, other evidence indicates that g-MDSCs are immature neutrophils with unique phenotypes [28,35,37]. Although N2 and g-MDSCs share similar features, recent evidence shows that they can be distinguished by surface markers, transcription profiles, and morphologies [36] as summarized in Table 1. Understanding the precise differences between N2 TANs and g-MDSCs using accurate nomenclature and surface marker expression is crucial for unraveling their specific roles in creating an immune-suppressive TME. Table 1 provides a summary of the characteristics of anti-tumor TANs (N1), pro-tumor TANs (N2), and PMN-MDSC/g-MDSCs in terms of their surface markers in mice and humans, polarization factors, and morphology. Establishing a standardized and universally accepted nomenclature for neutrophils is essential for making progress in understanding this complex immune cell type.

Numerous studies have consistently reported increased numbers of total neutrophils in the peripheral blood and tumor tissues of patients with different tumor types [28,35]. Patient NLR has been introduced as a prognostic factor for worse outcomes in NSCLC, colorectal cancer, breast cancer, melanoma, hepatocellular carcinoma, prostate cancer, ovarian cancer, cervical cancer, and renal carcinoma [14,19]. However, current clinical studies have primarily focused on measuring total circulating neutrophil counts, neglecting the roles of individual neutrophil subtypes. To gain a more comprehensive understanding of their impact on cancer outcomes, it is imperative that future clinical studies consider the measurement and analysis of distinct neutrophil subtypes. A recent work identified interferon-stimulated Ly6E^hi^ neutrophils in the blood as a potential biomarker for anti-PD-1 response in melanoma and lung cancer mice and patients [38]. Ly6E^hi^ neutrophils can also modulate the immune microenvironment and sensitize the tumor to anti-PD-1 treatment by activating CD8^+^ T-cells through IL-2b secretion [38]. With more studies like this, we can unravel the specific contributions of each subtype and improve our understanding of their significance in cancer prognosis.

**Table 1 cancers-16-02507-t001:** Summary of the characteristics of anti-tumor neutrophils (N1 TANs), pro-tumor neutrophils (N2 TANs), and polymorphonuclear myeloid-derived suppressor cells (PMN-MDSCs)/granulocytic myeloid-derived suppressor cells (g-MDSC). Commonly used standard surface markers are in bold, and additional markers are normal. Abbreviations: lymphocyte antigen 6 complex locus G6D (Ly6G), programmed death-ligand 1 (PD-L1), calprotectin (S100A8/A9), integrin alpha M (CD11b), sialic acid-binding Ig-like lectin 1 (CD170), neutrophil-specific antigen 1 (CD177), intercellular adhesion molecule 1 (CD54), Fc gamma receptor III (CD16), Fc gamma receptor II (CD32), Fc gamma receptor I (CD89), C-X-C motif chemokine receptor 2 (CXCR2), chitinase 3-like protein 3 (Ym1), gamma response 1 (GR1), SLAM family member 5 (CD84), colony-stimulating factor 1 receptor (CD115), natural killer cell receptor 2B4 (CD244), cluster of differentiation 14 (CD14), cluster of differentiation 15 (CD15), carcinoembryonic antigen-related cell adhesion molecule 8 (CD66), Lysyl oxidase (LOX), sialic acid-binding Ig-like lectin 3 (CD33), cluster of interleukin-4 receptor alpha chain (CD124), general control nonderepressible 2 (GCN2), reactive oxygen species (ROS), nitric oxide (NO), myeloperoxidase (MPO), hydrogen peroxide (H_2_O_2_), arginase 1 (ARG1), prostaglandin E2 (PGE_2_), metalloproteinase 9 (MMP9), epidermal growth factor (EGF), hepatocyte growth factor (HGF), platelet-derived growth factor (PDGF), calprotectin/alarmin (S100A8/A9), Lipopolysaccharide (LPS), interferon-gamma (IFNγ).

	Anti-Tumor Neutrophils (N1)	Pro-Tumor Neutrophils (N2)	PMN-MDSCs/g-MDSCs
**Surface markers in mice**	**CD11b^+^, Ly6G^+^, CD170^low^**, CD177^+^, CD54^+^, CD16^+^, CD32^+^, CD89^+^ [14,20,23,28]	**CD11b^+^, Ly6G^+^, Ly6C^low^, CD170^high^, PDL1^+^**, S100A8/A9^+^, CXCR2^+^, CXCR4^−^, Ym1^+^ [14,21,22,23,39,40]	**CD11b^+/low^, Ly6G^+^, GR1^+^, Ly6C^low^**, S100A8/A9^+^, CXCR4^+^, CD84^+^, CD115^+^, CD244^+^ [28,29,34,35,36]
**Surface markers in human**	**CD66b^+^, CD11b^+^, CD170^low^, CD66b^+^**, CD86^+^, CD54^+^, CD15^high^ [14,30]	**CD66b^+^, CD11b^+^, PDL1^+^, CD170^high^**, CD14^−^, CD15^+^, CXCR2^+^ [14,27,30]	**CD11b^+^, CD14^−^, CD15^+^, CD66b^+^**, LOX^+^, CD33^+^, CD84^+^, CD124^+^ [27,29,30,35,36,41]
**Polarization factors**	LPS, IFNγ [20,24,42,43]	TGFβ, IFNβ, IL-4 [23,43,44,45]	GCN2 [46]
**Factors released**	ROS, NO, MPO, H_2_O_2_ [14,17,20,47]	ROS, NO, ARG1, PGE_2_, MMP9, EGF, HGF, PDGF, NE, S100A8/A9 [14,20,45,48]	ROS, NO, ARG1, PGE_2_ [35,48]
**Morphology**	Mature, poly-nuclear, hyper-segmented, high density [23,27]	Mature, poly-nuclear, segmented nuclear, low density [23,27]	Mature or immature, ring-shaped nuclear, low density [41]
**Related cancer type**	Non-Hodgkin lymphoma, hepatocellular carcinoma, Lewis lung carcinoma, colon adenocarcinoma, renal cell carcinoma, breast cancer	NSCLC, hepatocellular carcinoma, melanoma, colon rectal cancer	Pancreatic ductal adenocarcinoma, NSCLC, breast cancer, renal cell carcinoma, head and neck squamous cell carcinoma, glioblastoma, melanoma

## 3. TAN Interaction with Malignant Cells

Tumors secrete granulocytic colony-stimulating factor (G-CSF), granulocytic-macrophage colony-stimulating factor (GM-CSF), and interleukin-17 (IL-17) to stimulate granulopoiesis in the bone marrow and neutrophil mobilization to blood vessels. This leads to an increase in both immature and mature neutrophils in the peripheral blood [49]. Tumors can also promote pro-tumor neutrophil recruitment and infiltration. In a mutant *Kras*-driven, *Trp53*-deleted NSCLC mouse model, lung adenocarcinoma activated osteocalcin-expressing osteoblastic cells in the bone marrow that remotely supply SiglecF^high^ TANs to promote tumor growth [39]. In NSCLC patients, the neutrophil level was elevated in the circulation, and neutrophil expansion was also detected locally within the tumor [11]. TAN expansion was associated with higher tumor burden and increased pro-tumor signaling pathways, which lead to poor clinical prognosis in NSCLC patients [50]. Here, we summarize factors released by neutrophils and how they affect malignant cells in NSCLC (Figure 1).

### 3.1. ROS

Reciprocal interactions between cancer cells and surrounding immune cells create chronic inflammatory TME. As fundamental components of the inflammatory response, neutrophils have both direct and indirect pro-tumor activities via multiple paracrine pathways [17]. Direct pro-tumor effects of neutrophils include ROS-induced genotoxicity, a boost in malignant cell proliferation through secreted cytokines, and senescence blockade through IL-1RA [17,18]. Releasing reactive oxygen species (ROS) was the first evidence that linked neutrophils with carcinogenesis. Neutrophil-derived ROS caused oxidative DNA damage in rat alveolar epithelial lung cells that induced base mispairing and mutations [51]. In a more recent study, ROS released by neutrophils caused DNA damage at the time of carcinogen exposure, leading to mutations that can induce tumorigenesis in lung cancers [52].

### 3.2. Growth Factors

Pro-tumor neutrophils secreted a variety of cytokines to sustain tumor growth, including epidermal growth factor (EGF), hepatocyte growth factor (HGF), and platelet-derived growth factor (PDGF) (Table 1) [53]. HGF in the tumor microenvironment contributes to tumorigenesis and progression in many human cancers. Pro-HGF is synthesized during myeloid cell maturation and stored in the granules of neutrophils. Neutrophils release the processed active HGF through degranulation. In lung adenocarcinoma, normal neutrophils were recruited from circulation to the tumor and induced to release active HGF to enhance malignant cell proliferation and migration [54].

In addition to releasing stored cytokines from the granules, neutrophils also synthesize prostaglandin E2 (PGE_2_) through cyclooxygenase (COX) from arachidonic acids. The overexpression of COX2, an isoform of COX, increased lung adenocarcinoma cell proliferation in vitro. When neutrophils co-cultured with lung adenocarcinoma cells, COX2-derived PGE_2_ secreted by neutrophils increased the proliferation of NSCLC cancer cells [55]. Elevated COX2 mRNA and protein levels were also associated with poor survival in vivo [56].

### 3.3. EMT Proteins

The indirect pro-tumor activities of neutrophils include enhancing epithelial–mesenchymal transition (EMT), the formation of neutrophil extracellular traps (NETs), the stimulation of angiogenesis, metabolic crosstalk, the inhibition of T-cell activity through immune checkpoint inhibition, and the activation of pro-tumor macrophages. In a *Kras*-driven lung cancer model, Faget and colleagues reported that TANs favored the KP tumor growth and sustained EMT transcription factor Snail in lung cancer cells. Snail-expressing tumor cells increased Cxcl2 secretion from neutrophils in the TME, thus accelerating pro-tumor neutrophil infiltration and tumor progression. The feedback loop between neutrophils and Snail-expressing cancer cells could form a vicious cycle, further enhancing immune suppression in the TME [57].

### 3.4. Neutrophil Extracellular Trap (NET)

Neutrophil extracellular trap (NET) is a unique feature of neutrophils that promotes metastasis in cancers [58]. NETs are primarily composed of antibacterial protein-coated DNA derived from neutrophils and extracellular fibers, first reported in 2004 [59]. They are released by activated neutrophils, triggered by inflammatory agents in response to pathogens, and serve as a part of innate immune defense against pathogens, trapping and killing microorganisms [59]. In the context of cancer, NETs catch circulating tumor cells (CTCs) from the peripheral blood in the premetastatic niche, stimulate the adhesion of CTCs to epithelial cells at the secondary tissue sites, and promote invasion and proliferation to form metastases [49,58]. Neutrophils from lung cancer patients are more likely to release NETs [60,61]. A high level of circulating NETs is also associated with unfavorable prognosis and might become a potential biomarker for progression and metastasis in lung cancer patients [60].

The NET-associated proteases neutrophil elastase (NE) and metalloproteinase 9 (MMP9) have also been reported to increase cancer cell proliferation. In breast cancer and prostate cancer mouse models with lung metastases, NET formation was promoted in lungs with sustained inflammation [62]. The NET-associated DNA binds to the ECM protein laminin and brings NE and MMP9 to initiate the cleavage of laminin. The proteolytic remodeling of laminin mediated by NET revealed an epitope that subsequently triggered the proliferation of dormant cancer cells and induced their proliferation through integrin-activated FAK/ERK/MLCK/YAP signaling [62].

MMP-9 is a marker for pro-tumor neutrophils that distinguishes them from anti-tumor neutrophils. It has a pivotal role in generating neovascularization. MMP9 turns on the “angiogenesis switch” by inducing the release of vascular endothelial growth factor (VEGF) in the ECM, resulting in expanded vascularization and tumor growth [63]. VEGF itself is also a potent tumor chemoattractant for neutrophils, regulated via IFN-β. The depletion of IFN-β leads to less migration of TANs, therefore leading to decreased angiogenesis and significantly reduced tumor growth [64]. In NSCLC, MMP-9 expression levels were highly associated with metastasis status and resulted in a reduced 5-year survival rate [65].

Neutrophil-derived polypeptide chemokine prokinectin 2 (PK2/BV8) has been demonstrated to have an important role in tumor angiogenesis. Anti-PK2/BV8 antibody treatment decreased the number of angiogenic islets and inhibited myeloid cell infiltration as well [66]. In human lung carcinoma, PK2/BV8 is predominantly associated with neutrophils, therefore indicating that PK2/BV8 from neutrophils might also contribute to the angiogenesis switch [53].

NET can also promote NSCLC metastasis by activating NF-κB and NOD-like receptor protein 3 (NLRP3) signaling pathways through long non-coding RNA MIR503HG [67]. Through a microarray analysis, it was reported that NETs released by the neutrophils induce the activation of NF-κB/NLRP3 inflammasome by downregulating MIR503HG expression, subsequently facilitating EMT and contributing to NSCLC metastasis. The overexpression of MIR503HR substantially inhibited the metastasis-promoting effect of NETs.

### 3.5. Chemokine Receptor

Neutrophils are produced in the bone marrow and released as mature and terminally differentiated cells, unlike other immune cells. Under normal conditions, circulating mature neutrophils make up a small fraction (1–2%) of the total neutrophil population in the body [68]. Neutrophils utilize chemokine receptors to respond to various chemokines. Neutrophils mainly express chemokine receptors from the CXC group, particularly CXCR1 and CXCR2 [69]. The retention of mature neutrophils in the bone marrow is regulated by the interplay between two chemokine receptors, CXCR4 and CXCR2 [70]. CXCR4^+^ neutrophils are anchored in the bone marrow by CXCL12 and secreted by osteoblasts and other stromal cells. Conversely, endothelial cells and megakaryocytes secrete CXCL1 and CXCL2, promoting the release of neutrophils into circulation via CXCR2 signaling [70,71,72]. High CXCR1 and CXCR2 expressions on neutrophils play an important role during their recruitment to the TME, in response to chemokines CXCL1, 2, 5, 6, and 8, secreted by malignant cells, fibroblasts, endothelial cells, and leukocytes [73,74]. CXCR2 activation triggers various signaling cascades, including PI3K/Akt for cell migration, PLC/PKC for cell function, and MAPK/p38 for cell proliferation and survival [75]. CXCR1 and CXCR2 signaling both activate the NF-kB pathway, leading to the transcription of cytokines and chemokines that enhance neutrophil recruitment [69].

In the mutant Kras-driven lung cancer mouse model, CXCR2-mediated and neutrophil-induced inflammation was associated with the malignant transformation of normal epithelial cells to lung adenocarcinoma. CXCR2 and its ligands were highly expressed in premalignant lesions in the lung and blocking CXCR2 inhibited the malignant progression [76]. In lung cancer patients, a significant association between elevated CXCR2 expression levels and prognosis has been reported. In lung cancer cell-challenged mice, increased CXCLs/CXCR2 signaling, Arg-1, and TGF-β levels were observed, accompanied by the significant infiltration of TANs in tumor tissues [77]. Targeting CXCR2 with a selective inhibitor promoted apoptosis, senescence, EMT, and suppressed cancer cell proliferation, which led to the pronounced inhibition of tumor growth. Blocking CXCR2 also resulted in the reduced infiltration of TANs, enhanced CD8^+^ T-cell activation, and the enhanced efficacy of traditional lung cancer chemotherapy like cisplatin [77].

### 3.6. Metabolites

A recent study reported interesting metabolic crosstalk between TANs and cancer cells that facilitates metastasis. The transcription profile of neutrophils in premetastatic lungs demonstrated overexpression in lipid droplet-associate genes compared to those in the peripheral blood [78]. Lung mesenchymal cells suppressed the activity of adipose triglyceride lipase (ATGL) in neutrophils, resulting in lipid accumulation. Those lipids were transported to cancer cells through the micropinocytosis–lysosome pathway to provide additional energy for metastasis in the lung of breast cancer mice [78].

### 3.7. Non-Coding RNAs

Non-coding RNAs refer to RNA molecules that, although not being translated into proteins, remain functional and can impact gene expression through diverse mechanisms. Regulatory non-long coding RNAs involved microRNAs (miRNAs), small interfering RNAs (siRNAs), piwi-interacting RNAs (piRNAs), and long non-coding RNAs (lncRNAs). MicroRNAs are composed of about 20 nucleotides and are capable of regulating downstream target gene expression and regulating immune cells in the TME [79]. Mir-223 in neutrophil is a suppressor of neutrophil maturation and miR223 knockout induces spontaneous lung inflammation in mice with excessive neutrophil infiltration and pro-inflammatory cytokines Cxcl2, Ccl3, and IL-6 [80]. Neutrophils can also deliver miR-223 into tumor cells upon direct contact through the gap junctions. In LUAD cells, miR-223 transferred from neutrophils to cancer cells initiated EMT, suppressed FOXO1, and increased migration and invasion [81].

On the other hand, lncRNAs are identified as RNAs larger than 200 nucleotides and they are incorporated into immune checkpoint blockade studies given their potential roles in regulating immune cell functions [82]. LINC01116 overexpression in the tumor recruited DDX5 to the interleukin-1β (IL-1β) promoter, increased IL-1β expression, and subsequently enhanced TAN recruitment [83]. The myeloid RNA regulator of Bim-induced death (Morrbid) is a lncRNA that controls neutrophil survival and lifespan [84]. Through the regulation of pro-apoptotic Bcl2l11 (Bim) transcription, Morrbid allowed the rapid control of the apoptosis of neutrophils. Further investigation of Morrbid upstream regulators and extracellular signals in the TME may provide promising approaches to target TANs. The lncRNA from the tumor has also been reported to influence the PD1/PD-L1 pathway in neutrophils, shedding light on a potential association with immunotherapy. Specifically, lncRNA HOXA transcript at the distal tip (Hottip) in ovarian cancer cells has been shown to enhance the transcription and secretion of interleukin-6 (IL-6) by binding to c-jun. IL-6 in turn stimulates the expression of PD-L1 on the neutrophil surface through STAT3 activation., resulting in immune evasion [85]. In a recently published study in TNBC, lncRNA metastasis-associated lung adenocarcinoma transcript 1 (Malat1) in malignant cells was reported to induce neutrophil infiltration by directly upregulating the Wnt signaling [86]. Malat1-KO or the ablation of the Wnt-activating enzyme Porcupine (PORCN) in the 4T 1 cells decreased TAN or PMN-MDSC infiltration and lung metastases while increasing CD8^+^ T-cells [86].

## 4. TAN Interactions with T-Cells

ICIs target receptors on T-cells to inhibit tumor progression. The function of T-cells is limited by programmed death 1 (PD-1) binding to its ligand programmed cell death ligand 1 (PD-L1) on the surface of tumor cells, leading to T-cell exhaustion. Anti-PD-1 antibodies like nivolumab and pembrolizumab bind to PD-1 on T-cells and restore the cytotoxic activity [87]. Similarly, the anti-CTLA-4 antibody blocks cytotoxic T-cell antigen 4 (CTLA-4) on the T-cells, prevents it from binding to its ligand B7 on the tumor cells, and therefore, turns off the inhibition of T-cell activity [88]. Although some studies showed neutrophils can present antigens and stimulate T-cell response, more evidence has demonstrated an immunosuppressive role of neutrophils in TME (Figure 1). In many circumstances, pro-tumor TANs negatively interfere with the efficacy of ICIs by interacting with T-cells in a dichotomous fashion [22]. A major hurdle in deciphering TAN-related immune suppression is the complex interaction of TANs with T-cells in the TME. Since cytotoxic T-cells are the major immune cells mediating antigen-driven anti-tumor effects, it is crucial to understand how TANs regulate T-cell activity that results in resistance to ICIs in cancer patients.

### 4.1. Neutrophil–Lymphocyte Ratio

In NSCLC, TANs were found to be the most represented CD45^+^ immune cell population. TAN abundance negatively correlated with tumor-reactive T-cell infiltration [11]. A meta-analysis study has suggested that a high neutrophil–lymphocyte ratio (NLR) (> or =4) could be a biomarker for poor prognosis in lung cancer patients [89]. The peripheral blood-derived neutrophil–lymphocyte ratio (NLR) has been used as a prognostic factor for overall survival and response to chemotherapy and ICIs in NSCLC [19]. High intratumoral TANs were associated with blunted T-cell response, characterized by decreased cytotoxic CD8+ T-cell markers—CD8A/B and GZMA/B—and IFNγ-related genes [50]. On the other hand, a recent study suggested that a low tumor-infiltrating neutrophil number increased IFNγ signaling and cytotoxic T-cells, thus enhancing the efficacy of anti-PD-(L)1 treatment in NSCLC patients [90]. This result was later confirmed with a study involving 221 NSCLC patients, showing that low NLR was linked to a more beneficial outcome in NSCLC patients treated with anti-PD-1 antibody pembrolizumab. Low NLR tumors also had a distinct TME, significantly associated with increased CD8^+^, PD-1^+^, and FOXP3^+^ cells [91]. NLR also has the potential as a blood test biomarker for immune-related adverse events (irAEs) after ICI treatment [92] but more studies need to be performed to confirm the predictive power of irAEs.

### 4.2. Direct Interaction with T-Cells

The expression of PD-L1 on the cell surface directly drives immune checkpoint engagement, impairs the anti-tumor immunity of T-cells, and leads to T-cell exhaustion. The upregulation of PD-L1 potentially causes PD-1-mediated T-cell apoptosis. Similar to malignant cells, TANs also express PD-L1, and several studies have suggested a correlation between neutrophil PD-L1 expression and immunosuppression in the TME [93]. A high frequency of PD-L1^+^Arg1^+^ neutrophils in the circulation was found in patients with NSCLC and correlated with poor prognosis [94]. The immune suppression of TANs via PD-L1 is interferon-dependent and requires direct cell–cell contact [48]. Blunted T-cell response in NSCLC patients was also correlated with a high PD-L1 expression on the intratumoral neutrophils [50]. PD-L1 expression can be regulated by IFNγ, and it has been implicated in the clinical response to ICI treatment. IFNγ signaling is an essential part of anti-tumor cytotoxicity. A strong inverse correlation has been observed between the IFNγ signature and the intratumoral TAN abundance [50]. In glioma mice models, the number of TANs also positively correlates with PD-L1 expression. Neutrophil depletion with ani-Lan anti-y6G antibody combined with anti-PD-1 treatment significantly inhibited tumor growth and improved survival outcome [95]. This translational study suggests that targeting TANs might increase the sensitivity to anti-PD-1 treatment and that the combination of neutrophil depletion with ICI therapies could potentially increase the efficacy of ICIs. However, the anti-Ly6G antibody does not only specifically target neutrophils but also recognizes g-MDSC. Additional studies are needed to determine if the depletion of tumor-associated neutrophils specifically contributes to sensitizing tumors to anti-PD-1 therapy.

### 4.3. Indirect Interaction with T-Cells

T-cell inhibition via pro-tumor TANs can also be established indirectly through the production of ROS and the release of arginase and through TAN degranulation and the release of cytokines. As discussed above, TAN-derived ROS promote tumor growth but also act on T-cells to abolish their anti-tumor activity. ROS can downregulate T-cell receptors on T-cells and, therefore, arrest them in the G0-G1 phase [48]. One of the reactive oxygen species, hydrogen peroxide (H_2_O_2_), induces T-cell apoptosis and decreases NF-κB activation to inhibit T-cell proliferation [22,48]. Of note, ROS production is not specific to pro-tumor TANs but is also a well-known anti-tumor function of neutrophils. ROS released by TAN in early-stage lung cancer exhibited anti-tumor activities. TANs within the tumor had more ROS production and phagocytic activity compared to circulating neutrophils [96]. Whether ROS is anti-tumor or pro-tumor might depend on the stage of the tumor, the status of neutrophil polarization, and T-cell infiltration.

Arginase 1 (ARG1) is released in response to G-CSF and TGFβ in the microenvironment and is also used as a marker for pro-tumor N2 neutrophils (Table 1). The overexpression of ARG1 by TANs decreased the availability of L-arginine in the TME, altering T-cell metabolism and resulting in T-cell dysfunction. In NSCLC patients, the ARG1^+^ neutrophil population increased with tumor stage and inversely correlated with the cytotoxic T-cell abundance [97]. ARG1 inhibitor treatment reduced tumor growth in *Kras*-driven NSCLC mice and increased both T-cell homing and activity [98].

CCL17, also known as thymus and activation-regulated chemokine (TARC), is highly expressed in pro-tumor N2 TANs and at a very low level in anti-tumor N1 TANs [23]. Upon the polarization of neutrophils via TGFβ, pro-tumor N2 phenotype showed a high expression of CCL17, ARG1, and CXCL14 [23]. In a xenotransplant lung cancer mouse model, CCL17 was indirectly upregulated via circulating RNA, resulting in decreased cytotoxic CD8^+^ T-cells and an increased T_regs_ [99]. Currently, there is no clinical study showing a correlation of CCL17 with the outcome of NSCLC patients but in other cancers, a high number of CCL17^+^ TANs within the tumor was correlated with a worse prognosis and tumor progression in hepatocellular carcinoma patients. A high expression of CCL17 on TANs was also associated with increased tumor size, immunosuppressive macrophages, and regulatory T-cell (T_regs_) migration [100]. CCL17 is known to induce the recruitment of T_regs_, specifically in cancer. T_regs_ are an immunosuppressive subset of CD4+ T-cells that play an important role in self-tolerance. The high tumor infiltration of T_regs_ is often associated with poor clinical outcomes in patients, through the inhibition of anti-tumor innate as well as adaptive immune responses. The depletion of T_regs_ in NSCLC mice diminished tumor burden and enhanced the recruitment of CD8^+^ T-cells [101]. The depletion of CCL17-secreting TANs also decreased the number of T_regs_ in the lung [102].

Neutrophils may also induce T_regs_ through the secretion of S100A8/A9 in a positive feedback manner. S100A8/A9, also called calprotectin, is very abundant in the cytosol of neutrophils and is involved in the inflammatory process [103]. The serum and tissue levels of S100A8/A9 could be a potential prognostic biomarker for the outcomes of patients undergoing immunotherapy. In NSCLC patients treated with anti-PD-1 antibody, Chao and colleagues reported that S100A8/S100A9 in peripheral blood were significantly increased among non-responders compared to responders [104]. S100A8/A9 induces a variety of immunosuppressive cells, thereby inducing T_regs_ de novo production and activation [105]. Blocking S100A8 in the mouse lung carcinogenesis model reduced the immunosuppressive cell function and restored T-cell anti-tumor activity [106]. S100A8/A9 induces a variety of immunosuppressive cells, thereby inducing T_regs_ de novo production and activation [105]. Blocking S100A8 in the mouse lung carcinogenesis model reduced the immunosuppressive cell function and restored T-cell anti-tumor activity [106].

The cytokine signaling between neutrophils and T-cells is bidirectional. TGFβ not only induces the polarization of N1 TANs to N2 TAN but also induces the differentiation of CD4+ Th17 cells. Th17 cells mainly secrete interleukin-17 (IL-17), which induces the activation and recruitment of neutrophils to the tumor [18]. In *Kras*-driven lung cancer mice, TAN recruitment was significantly increased while T-cell recruitment was reduced, due to IL-17 [107]. The presence of IL-17 also diminished the therapeutic effect of anti-PD-1 treatment. In lung cancer patients, a high level of IL-17 is correlated with a high neutrophil count, as well as lower T-cell numbers. The depletion of neutrophils or the blocking of IL-17 downstream signaling can sensitize the tumor to surmount anti-PD-1 resistance [107].

In recent years, other than their effect on cancer cells, NETs were also reported to interact with infiltrating T-cells and promote an immunosuppressive TME. In a mouse metastasis model with abundant NETs, tumor-infiltrating lymphocytes showed an exhausted phenotype and expressed inhibitory receptors. Treating the mice with DNase to target NETs could reduce tumor growth and NET formation, leading to an increased functional T-cell population [108]. PD-L1-expressing NETs were also elevated in patients with liver metastasis, and the administration of anti-PD-L1 can help restore functional T-cells and diminish tumor growth in patients [108].

In conclusion, tumor-associated neutrophils (TANs) play a complex role in regulating the T-cell population and activity in the tumor microenvironment. Meanwhile, TANs have been shown to negatively interfere with the efficacy of immune checkpoint inhibitors (ICIs) by inhibiting T-cell activity through various mechanisms, such as PD-L1 expression; ROS production; and the release of Arg, CCL17, S100A8/9, and NET formation. Studies have also suggested that targeting TANs may enhance the sensitivity of tumors to anti-PD-1 treatment. Further research is needed to better understand the intricate interaction between TANs and T-cells in the tumor microenvironment to develop effective strategies for overcoming TAN-related immune suppression and improving the efficacy of ICIs in cancer patients.

## 5. TAN Interaction with Other Immune Cells

S100A8 secreted by neutrophils can have multiple effects on the TME components (Figure 1). It has been reported that S100A8, but not S100A9, significantly induced PD-L1 expression in monocytes and macrophages in colon carcinoma mice [109]. By binding the TLR4 receptor, S100A8 induced an inflammatory response pathway and regulated PD-L1 expression via histone modification in monocytes and macrophages. The PD-L1 expression of the malignant cells was not affected by S100A8. Macrophages pre-exposed to S100A8 exhibited immunosuppressive phenotypes and suppressed T-cell proliferation and cytotoxicity [109]. Macrophage migratory activity increases in response to CCL2 and CCL17, which are cytokines most highly secreted by TANs. CCL2^+^CCL17^+^ TANs correlated with tumor size, differentiation, stage, vascular invasion, and poor prognosis in hepatocellular carcinoma patients. Neutrophil depletion enhanced the efficacy of sorafenib, a multi-kinase inhibitor, and decreased tumor growth [100].

Neutrophils have also been reported to regulate NK cell activities in breast cancer mouse models. In a tri-cell co-culture system with neutrophil NK cells and tumor cells, neutrophils potentially suppressed NK cell anti-tumor activity, even though neutrophils themselves exerted tumor toxicity [110]. Both anti-tumor and NK-cell inhibitory effects were mediated by ROS [110].

The interaction of TANs and B-cells may also offer new insights for regulating immunosuppression in the TME. The role of B-cells in tumors is still controversial. The anti-tumor function of B-cells is in part exerted through antibody-dependent cellular toxicity. In contrast, newly designated regulatory B-cells (B_regs_) can induce the differentiation of regulatory T-cells through anti-inflammatory IL-10 secretion, therefore inhibiting anti-tumor immune response [111]. Furthermore, B_regs_ can also promote the expression of PD-1 and PD-L1 to inhibit anti-tumor immunity in lung cancer [112]. Through STAT3 signaling, B_regs_ were also reported to support tumor growth and angiogenesis and to inhibit T-cell activity [113]. In a Lewis Lung Carcinoma mouse model, TANs increased the recruitment of B-cells via TNFα and modulated B-cell differentiation to CD148^+^ IgG-producing plasma cells [114]. Neutrophils were also reported to inhibit B-cell activity and induce cell death by releasing ROS, NO, and ARG1 [115,116].

In summary, neutrophils have diverse effects on the components of the tumor microenvironment (TME), including inducing PD-L1 expression in monocytes and macrophages, regulating macrophage immunosuppressive phenotypes, modulating NK cell anti-tumor activity through ROS, and influencing B-cell recruitment and differentiation. Further understanding of the intricate interplay between neutrophils and other immune cells in the TME could provide novel insights for developing strategies to regulate immunosuppression.

## 6. Targeting Pro-Tumor Neutrophils and Clinical Trials That Combined with ICIs

Tumor-associated neutrophils are one of the major barriers that affect the efficacy of immune checkpoint blockade treatment. Targeting TANs or TAN-secreted factors in combination with ICIs might provide benefits to overcome resistance to ICI treatment. Current approaches that target neutrophils include (1) neutrophil depletion, (2) blocking neutrophil recruitment and polarization, (3) inhibiting pro-tumor factors secretion from neutrophils, and (4) reducing immune suppression in the TME [117] (Figure 2). Recent publications have summarized the ongoing clinical trials of agents with putative effects on neutrophils [19,20] and targeting neutrophils in association with ICI treatment [22,118]. Here, by applying the predefined search items of “neutrophil” and “immunotherapy”, we found 129 clinical trials on https://ClinicalTrials.gov (accessed on 28 November 2023). Of these, 32 were excluded due to unknown results or terminated statutes, and another 3 were added based on the new literature for a total of 21 compounds. A total of 97 clinical trials were considered for this review, of which 49 had been completed and had available clinical results. Drugs that potentially affect neutrophil recruitment, polarization, signaling, or protein secretion are currently being tested in combination with ICIs for NSCLC patients (Table 2).

Neutrophil depletion with anti-Ly6G antibody greatly enhanced the therapeutic effect of anti-PD-1 treatment in preclinical models [95] but has not yet entered the clinical trial. Universal neutrophil depletion might not be well tolerated in cancer patients. The majority of the drugs in Table 2 target secreted factors, like ROS, CCL2, NE, IL6, MMPs, and VEGFs from neutrophils. These approaches aim to block the crosstalk between pro-tumor neutrophils and malignant cells and between immunosuppressive neutrophils and cytotoxic T-cells. Such treatment has shown enhanced T-cell-dependent cytotoxicity [118] and could be a promising strategy to increase the sensitivity to ICI treatment.

Targeting the interaction between pro-tumor TANs and T_regs_ could be beneficial in combination with ICI treatment. Tasquinimod, which inhibits S100A9, has been approved to treat prostate cancer as a monotherapy [119]. Since the S100A9 serum level was particularly high in anti-PD-1 non-responders in NSCLC, Tasquinimod combined with anti-PD-1 antibody might be a promising therapy to investigate further.

In summary, targeting TANs or TAN-secreted factors in combination with ICIs, such as through neutrophil depletion, blocking neutrophil recruitment and polarization, or inhibiting pro-tumor factor secretion from neutrophils, could potentially overcome resistance to ICI treatment and enhance anti-tumor immunity. Ongoing clinical trials are investigating the use of agents that target neutrophils in combination with ICIs, and promising results have been shown.

**Table 2 cancers-16-02507-t002:** Clinical trials of agents with putative effects on neutrophils in cancer patients as accessed using records identified from https://clinicaltrials.gov on 8 March 2023. Trials in NSCLC and in combination with ICI treatments were included here, and if there are no current trials in NSCLC, other disease types were included instead. Abbreviations: NA, not applicable; transforming growth factor beta necrosis factor alpha (TGF-β); granulocyte colony-stimulating factor (G-CSF); colony-stimulating factor 1 receptor (CSFR1); phosphodiesterase 5 (PDE5); cyclooxygenase 2 (COX2); C-X-C motif chemokine receptor 2/5 (CXCR2/5); interleukin 6 (IL-6); Toll-like receptor 9 (TLR9); vascular endothelial growth factor receptor (VEGFR); platelet-derived growth factor receptor (PDGFR); metalloproteinase (MMP); arginase 1 (ARG1); peptidylarginine deiminase 4 (PAD4); S100 calcium-binding protein A9 (S100A9); lymphocyte antigen 6 complex locus G6D (Ly6G); C-C motif chemokine ligand 2; neutrophil elastase (NE); and reactive oxygen species (ROS).

Drug	Target	Disease	Status	NCT	Combination
Galunisertib	TGFβ	NSCLC	Phase II	NCT02423343	Nivolumab [120]
Pegfilgrastim	G-CSF	NSCLC	Phase I	NCT01840579	Pembrolizumab and chemo
Cabiralizumab	CSF1R	NSCLC	Phase I	NCT03502330	Nivolumab
Sidenafil	PDE5	NSCLC	Phase III	NCT00752115	Carboplatin, paclitaxel
Tadalafil	PDE5	NSCLC	Terminated at phase II	NCT04069936	Nivolumab
Celecoxib	COX2	NSCLC	Phase II	NCT00030407	Docetaxel
BMS-813160	CXCR2/5	NSCLC	Phase II	NCT04123379	Nivolumab
Navarixin	CXCR2	NSCLC	Phase II	NCT03473925	Pembrolizumab
Reparixin	CXCR2	Metastatic breast cancer	Phase I	NCT02370238	paclitaxel
Telaglenastat	glutaminase	NSCLC	Terminated at phase II	NCT04265534	Pembrolizumab and chemo
Tocilizumab	IL6	NSCLC	Phase II	NCT04691817	Atezolizumab
DV281	TLR9	NSCLC	Phase I	NCT03326752	Nivolumab
Anlotinib	VEGFR, PDGFR	NSCLC	Phase II	NCT05001971	Penpulimab [121]
Marimastat	MMP	NSCLC	Phase III	NCT00002911	NA
Numidargistat	ARG1	Metastatic/solid tumor	Phase II	NCT02903914	Pembrolizumab
GSK484	PAD4, NETosis	Lung carcinoma	Preclinical [122]	NA	Anti-PD1, anti-CTLA4
Tasquinimod	S100A9	Prostate cancer	Approved [119]	NCT01234311	NA
1A8	Ly6G	Glioma	Preclinical [95]	NA	Anti-PD1
Bindarit	CCL2	Diabetic nephropathy	Phase II	NCT01109212	NA
Sivelestat	NE	Acute lung injury in esophageal cancer	Approved in Japan and Korea [123]	NCT01170845	NA
Bardoxolone methyl	ROS	Advanced solid tumor	Phase I	NCT00529438	NA

## 7. Conclusions

TANs in the tumor microenvironment have multifaceted roles and have become increasingly recognized as an important component in lung cancer. Due to the heterogeneity of neutrophils and similarities with g-MDSCs, a nomenclature for specific neutrophil subsets must be used with caution to avoid ambiguity and confusion. An analysis of TANs and associated factors in the peripheral blood of patients has revealed poor prognosis and responses to immune checkpoint inhibitors. TANs promote tumor progression, angiogenesis, and metastasis by releasing ROS, cytokines, metabolites, and NETs. Neutrophils also contribute to an immunosuppressive TME through the inhibition of cytotoxic T-cells, B-cells, and NK cells while promoting regulatory T-cells. Elevated pro-tumor neutrophils and the release of immunosuppressive cytokines potentially lead to resistance to immune checkpoint blockade treatment. Many studies show a correlation between elevated TAN abundance/activity and resistance to ICI; however, the mechanisms still have multiple gaps. The convoluted interactions of TANs with malignant cells, T-cells, and other TME components require further elucidation in NSCLC. The majority of the current clinical trials of combination therapy are in advanced metastatic tumors with very few focusing on NSCLC. For neutrophils, it will be important to understand if neutrophil depletion or targeting neutrophil-secreted factors will be a promising therapeutic approach for lung cancer patients. For ICI non-responders, targeting neutrophils in combination with ICIs may potentially enhance the efficacy of ICIs and improve patient outcomes. A further study of the recruitment response, the immune exclusion, and the molecular mechanisms of TAN immunosuppression in NSCLC is still required.

## Figures and Tables

**Figure 1 cancers-16-02507-f001:**
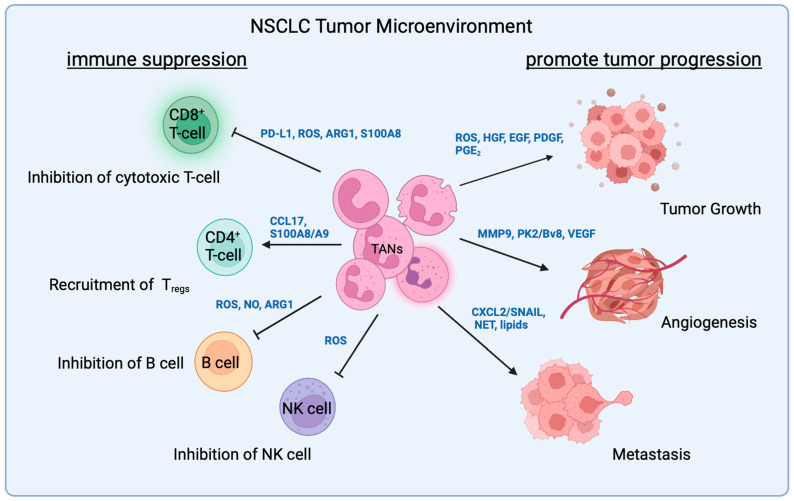
Functions of pro-tumor neutrophils in the TME. 0TANs secret chemokine, cytokines, and other factors that lead to tumor growth, including ROS, HGF, EGF, PDFF, and PGE_2_. TAN-released MMP9, PK2/Bv8, and VEGF also contribute to angiogenesis. TAN can also facilitate metastasis through CXCL2/SNAIL signaling, the formation of NET, and altering lipid metabolism. TANs also contribute to immune suppression through the inhibition of cytotoxic T-cells via PD-L1, ROS, ARG1, and S100a8/A9. They also recruit regulatory T-cells through CCL17 and S100A8/A9. ROS, NO, and ARG1 secreted by TANs can suppress B-cell activity and ROS may lead to the inhibition of NK cell’s anti-tumor activity. Abbreviations: programmed death-ligand 1 (PD-L1), reactive oxygen species (ROS), epidermal growth factor (EGF), hepatocyte growth factor (HGF), arginase 1 (ARG1), S100 calcium-binding protein A8 (S100A8), calprotectin (S100A8/A9), C-C motif chemokine ligand 17 (CCL17), vascular endothelial growth factor (VEGF), metalloproteinase 9 (MMP9), prokineticin 2 (PK2/Bv8), nitric oxide (NO), C-X-C motif chemokine ligand 2 (CXCL2), zinc-finger protein SNAI1 (SNAIL), neutrophil extracellular net (NET), and prostaglandin E2 (PGE_2_).

**Figure 2 cancers-16-02507-f002:**
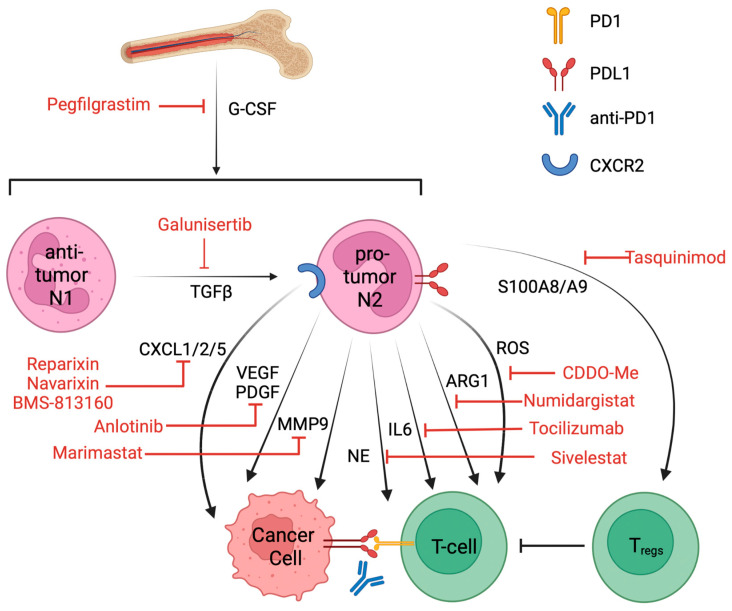
Inhibition of TANs in association with ICI treatment potentially enhances ICI efficacy. Pegfilgrastim targets G-CSF and blocks neutrophil recruitment from the bone marrow. Galuniserib targets TGFβ and potentially inhibits pro-tumor N2 polarization; Reparixin, Navarixin, and BMS-813160 target CXCL1/2/5 chemotaxis of neutrophil; Anlotinib blocks VEGF and PDGF; Marimastat targets MMP 9; and they all aim to inhibit the pro-tumor effect of neutrophils. Sivelestat targets NE; Tocilizumab targets IL-6; Numidargistat targets ARG1; CDDO-Me blocks ROS; and they all aim to reverse the inhibition of cytotoxic T-cells by neutrophils. Tasquinimod targets S100A8/A9 and inhibits the recruitment of regulatory T-cells. Abbreviations: granulocyte colony-stimulating factor (G-CSF), transforming growth factor beta necrosis factor alpha (TGF-β), C-X-C motif chemokine ligand 1/2/5 (CXCL1/2/5), vascular endothelial growth factor (VEGF), platelet-derived growth factor (PDGF), metalloproteinase 9 (MMP9), neutrophil elastase (NE), interleukin 6 (IL-6), arginase 1 (ARG1), reactive oxygen species (ROS), and S100 calcium-binding protein A8/A9 (S100A8/A9).

## Data Availability

No new data were created or analyzed in this study, all data and references can be found on https://pubmed.ncbi.nlm.nih.gov.

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
