# Peer review of "The Multifaceted Role of Neutrophils in NSCLC in the Era of Immune Checkpoint Inhibitors"

_cancers, 2024, doi:10.3390/cancers16142507_

Round 1

Reviewer 1 Report

Comments and Suggestions for Authors

I agree with publication of this manuscript. Author highlight the importance of tumor-associated neutrophils (TANs) in non-small cell lung cancer (NSCLC). Effects of TANs in the NSCLC TME and their link to immune are well articulated. The review's focus on neutrophils is likely to stimulate further research and potentially lead to breakthroughs in overcoming ICI resistance. It would be better to provide some more NET related research if possible.

Author Response

Point 1: We thank the reviewers for their encouraging words on our review paper and appreciate the suggestion regarding the role of neutrophil extracellular traps (NETs) in NSCLC. In Section 3.4 between Lines 225-227, we added two references “Zhang et al. BioMed Research International 2022 (Ref 60)” and “Mauracher et al. Research&Practice in Thrombosis and Hemostasis, 2023 (Ref 61)” to discuss NET in lung cancer patients. In Section 3.4 between Lines 251-256, we added a reference “Wang et al., Frontiers in Immunology 2022 (Ref 67)”, to describe how NET induces NSCLC metastasis through regulating lncRNA MIR503HG.

Point2: We also included the discussion regarding the role of NET in immunosuppressing T-cell activity as highlighted in the study by Kaltenmeier et al., Frontier Immunology 2021 (Ref 108), in Section 4.3 between Lines 439-446. This is a fascinating concept that holds significant implications for understanding the interplay between NET and T-cells in the context of immunotherapy response and we really appreciate your suggestion.

Reviewer 2 Report

Comments and Suggestions for Authors

In this review,  the authors summarize current studies of neutrophil interaction with malignant cells, T-cells, and other components in the TME.

I suggest the authors summarize neutrophils as biomarkers for ICI, if available, please meta-analysis?

The subtypes of neutrophils, and dynamic change of neutrophil?

and neutrophil for fast progression or irAE (https://www.frontiersin.org/journals/immunology/articles/10.3389/fimmu.2022.862752/full)

Author Response

Point 1: Thank you for pointing out the importance of neutrophils as biomarkers for ICI response. We added a very recent publication “Benguigui et al. Cancer Cell, 2024 (Ref 38)” to elaborate on the role of interferon-stimulated neutrophils as a predictor of ICI response in Section 2, Lines 140-146. Unfortunately, we do not have the capacity to perform a meta-analysis at this point but we have cited a meta-analysis work on NLR as a biomarker of lung cancer patient outcomes “Yu et al. Molecular and Clinical Oncology, 2017 (Ref 89)”.

Point 2: We have discussed the neutrophil subtypes and heterogeneity in Section 2, starting from Line 71. We also highlighted the dynamic change of neutrophils to “N1” and “N2” phenotypes in the same section, starting from Line 91.  

Point 3: We agree with the inclusion of immune-related adverse events (irAEs) as highlighted in the suggested study by Zhou et al., Frontier Immunology 2022. In light of your suggestion, we added a brief discussion in Section 4.1 from lines 354 to 361.

Reviewer 3 Report

Comments and Suggestions for Authors

Overall, this is an excellent review article.

A minor comment: Line 458-459, please cite a review article related to the pro-tumour role of B cells in cancer.

Author Response

Thank you We appreciate the reviewer bringing this important point to our attention and we now included a couple of review papers like “Sarvaira et al. Cellular and Molecular Immunology, 2017 (Ref 111)” and “Leong et al. Translational Lung Cancer Research, 2021 (Ref 112)” in Section 5 between Line 479-484 of the pro-tumor roles of B cells in lung cancer.